Fine-scale mapping of chromosome 9q22.33 identifies candidate causal variant in ovarian cancer

Xing Tongyu
Zhao Yanrui
Wang Lili
Geng Wei
Liu Wei
Zhou Jingjing
Huang Caiyun
Wang Wei
Chu Xinlei
Liu Ben
Chen Kexin
Zheng Hong zhengh64@aliyun.com
Li Lian lilian@tmu.edu.cn
Department of Epidemiology and Biostatistics, Key Laboratory of Molecular Cancer Epidemiology, Key Laboratory of Prevention and Control of Human Major Diseases, Ministry of Education, Tianjin’s Clinical Research Center for Cancer, National Clinical Research Center for Cancer, Tianjin Medical University Cancer Institute and Hospital, Tianjin Medical University , Tianjin , China
Coates Philip
Electronic publication date: 2024 Feb 14
Publication date: 2024
Volume: 12
Electronic Location ID: e16918
Received 2023 Sep 15; Accepted 2024 Jan 18
Copyright: ©2024 Xing et al.
Copyright year: 2024
Copyright holder: Xing et al.
License: This is an open access article distributed under the terms of the Creative Commons Attribution License, which permits unrestricted use, distribution, reproduction and adaptation in any medium and for any purpose provided that it is properly attributed. For attribution, the original author(s), title, publication source (PeerJ) and either DOI or URL of the article must be cited.
License URL: https://creativecommons.org/licenses/by/4.0/

Keywords: Fine-scale mapping, Ovarian cancer, Genetic variants, Functional annotation, Expression quantitative trait locus analysis

Funding: National Key R&D Program of China 2021YFC2500400 National Natural Science Foundation of China 81973113 National Human Genetic Resources Sharing Service Platform 2005DKA21300 The National Key Research and Development Program of China The Net Construction of Human Genetic Resource Bio-bank in North China 2016YFC1201703 Tianjin Key Medical Discipline (Specialty) Construction Project TJYXZDXK-009A This work was supported by the National Key R&D Program of China (2021YFC2500400), the National Natural Science Foundation of China (81973113), the National Human Genetic Resources Sharing Service Platform (2005DKA21300), the National Key Research and Development Program of China, the Net Construction of Human Genetic Resource Bio-bank in North China (2016YFC1201703), and the Tianjin Key Medical Discipline (Specialty) Construction Project (TJYXZDXK-009A). The funders had no role in study design, data collection and analysis, decision to publish, or preparation of the manuscript.

==============================
Ovarian cancer is a complex polygenic disease in which genetic factors play a significant role in disease etiology. A genome-wide association study (GWAS) identified a novel variant on chromosome 9q22.33 as a susceptibility locus for epithelial ovarian cancer (EOC) in the Han Chinese population. However, the underlying mechanism of this genomic region remained unknown. In this study, we conducted a fine-mapping analysis of 130 kb regions, including 1,039 variants in 200 healthy women. Ten variants were selected to evaluate the association with EOC risk in 1,099 EOC cases and 1,591 controls. We identified two variants that were significantly associated with ovarian cancer risk (rs7027650, P = 1.91 × 10−7; rs1889268, P = 3.71 × 10−2). Expression quantitative trait locus (eQTL) analysis found that rs7027650 was significantly correlated with COL15A1 gene expression (P = 0.009). The Luciferase reporter gene assay confirmed that rs7027650 could interact with the promoter region of COL15A1, reducing its activity. An electrophoretic mobility shift assay (EMSA) showed the allele-specific binding capacity of rs7027650. These findings revealed that rs7027650 could be a potential causal variant at 9q22.33 region and may regulate the expression level of COL15A1. This study offered insight into the molecular mechanism behind a potential causal variant that affects the risk of ovarian cancer.

Introduction

Ovarian cancer is one of the most common gynecological cancers and is the second most fatal gynecological malignancy in Chinese women (Sung et al., 2021). As a complex polygenic disease (Dareng et al., 2022; Stewart, Ralyea & Lockwood, 2019), genetic factors play a significant role in ovarian cancer etiology (Jones et al., 2017). So far, over 40 susceptibility loci of ovarian cancer have been identified by genome wide association studies (GWAS) (Glubb et al., 2020; Kar et al., 2018; Lawrenson et al., 2019; Manichaikul et al., 2020). However, the mechanisms through which the most of these loci contribute to the ovarian cancer carcinogenesis remain unclear. A functional study was conducted at 9p22.2 locus, which was the first and most significant risk locus for ovarian cancer identified through GWAS in European women. Buckley et al. (2019) identified multiple candidate causal variants (CCVs) at the locus and these variants may be mediated by changes in a transcriptional regulatory network of several regulatory elements that act on BNC2, potentially influencing ovarian cancer susceptibility.

In our previous GWAS, we determined that rs1413299 at 9q22.33 was significantly associated with the risk of epithelial ovarian cancer (EOC) in Chinese women (Chen et al., 2014). SNP rs1413299 is located at intron six of COL15A1, which encodes the alpha chain of collagen type XV (COL15). COL15, a member of the FACIT (fibril-associated collagens with interrupted triple helices) family, is a proteoglycan closely associated with the basement membrane, which has a broad tissue distribution (Bretaud et al., 2020; Clementz & Harris, 2013). COL15A1 gene expression levels were significantly different between ovarian cancer tumor tissues and normal tissues in The Cancer Genome Atlas (TCGA) data (Chen et al., 2014).

However, the effect of the variants in this region on the occurrence and development of ovarian cancer is still unclear. To identify the potential causal variants and additional novel variants at 9q22.33, we conducted a target sequencing followed by a validation study in a large sample size. We also performed a preliminary functional exploration to investigate the molecular mechanism of the susceptibility region.

Materials and Methods

Study subjects were recruited from the Tianjin and Guangdong provinces in China. Cases were newly diagnosed and histologically confirmed to be epithelial ovarian cancer. Controls were selected from healthy and cancer free women who underwent regular health check-ups or participants in the cancer screening program. Informed consent was obtained from each subject when recruited. This study was approved by the Medical Ethics Committee of Tianjin Medical University Cancer Institute and Hospital (Approval number: bc201810). 200 cancer free women were selected for target sequencing and a total of 1,099 epithelial ovarian cases and 1,591 age matched controls were included in the association study.

Target region sequencing and genotyping

We explored linkage disequilibrium (LD) structure around the target SNP rs1413299 using the HapMap Project database (Release 3 version 27, CHB+JPT) by Haploview (Barrett et al., 2005). A total of 1,039 variants (total bases: 130 kb, overall coverage: 96.9%) were successfully designed within the LD block interval at 9q22.33 (chr9:101,730,000-101,860,000, hg19). Sequencing was conducted using Ion Torrent PGM Next Generation Sequencing (Thermo Fischer Scientific, Waltham, MA, USA) and four variants with minor allele frequency (MAF) >5% and in high LD with the target SNP rs1413299 (r2>0.8) were selected. In order to identify additional new variants, we also included SNPs with P < 1 ×10−3 in the 1Mb region surrounding rs1413299 from previous GWAS results and the SNPs with high LD (r2>0.7) to rs1413299 from SNAP (https://www.broadinstitute.org/snap/snap). Finally, ten variants were included for the subsequence validation study.

Genotyping of the ten SNPs (rs10819587, rs10988451, rs1413298, rs1572136, rs1889268, rs4743305, rs7021675, rs7027650, rs7031588, and rs73503719) was performed using the iPlex MassARRAY platform (Sequenom, Inc.) according to manufacturer’s protocols.

Functional annotation

The functional annotations of the variants were performed using several kinds of sources. Epigenomic data was obtained from the Roadmap Epigenomic project (https://egg2.wustl.edu/roadmap/data/byFileType/signal/consolidated/macs2signal/foldChange/), including DNase-Seq data and Chip-seq profiles for H3K27ac and H3K4me1 histone modification of primary ovary tissue E097. The ATAC-seq profiles for three ovarian cancer cell lines (OVCA432, DOV13, and SKOV3) were from our previous study (Dai et al., 2021). The construction method of the ATAC-Seq library and analysis method has been described previously (Dai et al., 2021). We also used HaploRegv4.1 (https://pubs.broadinstitute.org/mammals/haploreg/haploreg.php) (Ward & Kellis, 2012) and RegulomeDB (http://regulomedb.org/) (Boyle et al., 2012) to explore the possible functions of the variants.

eQTL analysis

We obtained 102 ovarian tumor tissues from the cancer biobank of Tianjin Medical University Cancer Hospital and extracted the total RNA from these tissues using the standard Trizol method. Real-time PCR was used to measure COL15A1 gene expression. The eQTL analysis was conducted using linear regression, performed with the R package Matrix eQTL, and adjusted for age.

Cell lines

The SKOV3, 293T, and OVCA433 cell lines were preserved in the Laboratory of Epidemiology and Biostatistics at Tianjin Medical University Cancer Hospital. SKOV3 was cultured in RPMI 1640 medium with 10% FBS. 293T and OVCA433 were cultured in DMEM medium with 10% FBS, and incubated in a cell incubator at 37 °C with 5% CO2.

Luciferase assay

We downloaded the target DNA fragment containing rs7027650 from the NCBI (https://www.ncbi.nlm.nih.gov/), and designed the upstream and downstream primers of the target fragment. PCR amplification was performed using the whole genome DNA of the ovarian cancer cell line SKOV3 as a template to obtain a 500 bp DNA fragment containing the SNP rs7027650. Using the PGL3 Basic-COL15A1 Promoter as a control, we inserted DNA fragment-A and DNA fragment-T into the PGL3 Basic-COL15A1 Promoter plasmid vector to obtain PGL3 Basic-A-COL15A1 Promoter and PGL3 Basic-T-COL15A1 Promoter plasmid vector. The three plasmids were then transfected into SKOV3 ovarian cancer cell line or 293T cell line, respectively, and luciferase activities were detected after 30 h using the Dual Luciferase Assay Kit (Promega, Madison, WI, USA).

Electrophoretic mobility shift assay (EMSA)

The probes were commercially synthesized and labeled with biotin at the 5′  end (Table S1), and nuclear protein was obtained from ovarian cancer cell line OVCA433. EMSA assays were performed using the LightShift Chemiluminescent EMSAs Kit (Thermo Fisher Scientific) according to the manufacturer’s instructions and as described previously (Dai et al., 2021).

Statistical analysis

The association between the variants and ovarian cancer risk was estimated using logistic regression adjusted for age. Linkage disequilibrium between the variants and lead SNP rs1413299 was evaluated by PLINK 1.9. The figure was made using R software version 4.0.4 (R Core Team, 2021) and the difference was estimated using a Student’s t-test. The P values were two-sided, and P<0.05 was considered statistically significant.

Results

Targeted sequencing of the 9q22.33 region

Variants for fine-mapping were selected by Haploview software based on the Asian population (CHB+JPT) data from Hapmap Release3 version 27. We obtained 1,039 variants at 130 kb (Chr9: 101,730,000–101,860,000) (Fig. S1) of the 9q22.33 region and performed targeted sequencing in 200 healthy women. We identified four variants that have high LD (r2 > 0.70) with rs1413299 and MAF >0.05 in target sequencing. We also selected six variants according to the results of the previous GWAS (Chen et al., 2014) and analyzed these variants with SNAP (details in ‘Materials and Methods’). In total, ten variants were available for genotyping using Sequenom MassArray and were used for subsequent association analysis (Tables S2–S4 and ‘Materials and Methods’).

Association analysis of the variants and ovarian cancer risk

Ten variants were genotyped in 1,099 EOC cases and 1,591 healthy controls (Table S5) using the Sequenom iPlex platforms. We identified that rs7027650 (P = 1.91 × 10−7, OR =1.31, 95%CI [1.21−1.41]) and rs1889268 (P = 3.71 × 10−2, OR =1.14, 95%CI [1.02−1.26]) were significantly associated with ovarian cancer risk in additive model (Table 1 and Table S6). The association between rs7027650 and ovarian cancer risk was also significant in both the dominant (P = 5.90 × 10−10, OR =1.80, 95%CI [1.62−1.98]) and recessive models (P = 4.88 × 10−3, OR =1.26, 95%CI [1.10−1.42]). For rs1889268, a significant association was also observed in recessive model (P = 2.32 × 10−2, OR =1.20, 95%CI [1.04−1.35]). We then performed stratification analysis by histological subtype, finding that rs7027650 was significantly associated with three major histotypes of epithelial ovarian cancer, (Serous OC: P = 1.33 × 10−3; Endometrioid OC: P = 5.57 × 10−6; Mucinous OC: P = 1.27 × 10−3). Additionally, we found that rs1889268 was significantly associated with Serous OC (P = 4.99 × 10−2), and rs1413298 was associated with Endometrioid OC (P = 8.72 × 10−3) (Fig. 1 and Table S7)

Table 1 Association results of SNPs at 9q22.33 in validation stage.

SNP	Positiona	Alleleb	RAF	Genotypes	Case/Control	OR (95% CI)	P	
rs7027650	101,741,969	T/A	0.57	AA	182/422	1.00(reference)		
				TA	450/569	1.82 (1.47–2.26)	3.44E−08	
				TT	419/522	1.86 (1.50–2.31)	1.69E−08	
				Addictive		1.39(1.24–1.55)	1.91E−07	
				Dominant		1.80(1.62–1.98)	5.90E−10	
				Recessive		1.26(1.10–1.42)	4.88E−03	
rs1889268	101,767,961	T/C	0.32	CC	468/755	1.00 (reference)		
				TC	488/656	1.20 (1.02–1.41)	3.16E−02	
				TT	115/153	1.21 (0.93–1.58)	1.61E−01	
				Addictive		1.13(1.02–1.25)	3.71E−02	
				Dominant		1.11(0.85–1.37)	4.23E−01	
				Recessive		1.20(1.04–1.35)	2.32E−02	
Notes.

CI confidence interval

OR odds ratio

RAF risk allele frequency

SNP single-nucleotide polymorphism

a Position is given with respect to genome build 37.

b Risk allele/other allele.

Figure 1 The association between three variants and EOC risk stratified by histological subtype.

ORs (95% CI) and P values for associations of three variants with each of the histological subtypes of EOC risk which were estimated using logistic regression adjusted for age.

Functional annotation and eQTL analysis of candidate causal variants

We annotated these SNPs using publicly available data from the Roadmap project (DNase-Seq data and Chip-seq profiles for H3K27ac and H3K4me1 histone modification of primary ovary tissue) and in-house ATAC-seq profiles in three ovarian cancer cell lines (OVCA432, DOV13, and SKOV3). We found that rs7027650, the most significant variant, overlapped with DNase hypersensitivity peaks in normal primary ovarian tissue which represent the open chromosome region. ATAC-seq analyses showed that rs1889268 and rs1413298 resided in open chromatin regions in ovarian cancer cells (Fig. 2A).

Figure 2 Epigenetic annotations and eQTL analysis for three candidate variants.

(A) Functional annotations with epigenomics data of normal ovary tissue and ovarian cancer cell lines. The lines represent H3K27ac and H3K4me1 histone modification ChIP-seq profiles, DNase-Seq for primary ovary tissue, and ATAC-seq profiles for ovarian cancer cell lines (OVCA432, DOV13, and SKOV3) from top to bottom. The region of rs7027650 overlaps with the peak of the DNase I hypersensitivity site measured by DNase-Seq. (B) Results of eQTL analysis in 102 ovarian cancer tumor tissues showed the significant association between rs7027650 and mRNA expression of COL15A1.

To identify candidate functional variants, we then performed expression quantitative trait locus (eQTL) analysis in 102 ovarian tumor tissues from Chinese women. We found that the allele T of rs7027650 was significantly associated with the decreased mRNA expression of COL15A1 (P = 0.009) (Fig. 2B and Fig. S2). However, there was no significant association between the other two variants and expression of the COL15A1 gene (rs1889268: P = 0.072; rs1413298: P = 0.193). We have also explored the GTEx project data, we found that there were several variants associated with gene expression. The most significant variant, 7027650, was associated with COL15A1 expression (P <  0.05) in Thyroid and Nerve-Tibial tissues. However, the significant correlation was not observed in normal ovary tissues (Table S8).

Validation of the candidate causal variants

We performed the luciferase assay and EMSA to explore the possible function of potential variants. In order to explore whether the variant interacts with the promoter of COL15A1, we constructed luciferase reporter vectors containing the rs7027650-A allele or rs7027650-T allele and transfected these plasmids into the SKOV3 and 293T cell lines, respectively. Compared to the fragment containing allele-A, the transcription activity of COL15A1 in the fragment with allele-T was lower in both SKOV3 and 293T cells (Fig. 3A). The results suggested that the gene fragment containing rs7027650 could interact with the promoter region of COL15A1 and may reduce its activity. Next, we performed EMSAs using the probes containing allele A, allele T, and controls. The results showed differences in the allele-specific protein binding capacity of rs7027650 (Fig. 3B). These results indicate that rs7027650 may be a potential functional SNP.

Figure 3 Functional validation of rs7027650.

(A) Luciferase reporter assay using vector containing rs7027650 in SKOV3 and 293T cells. The Luciferase reporter assay showed the different activity of effect allele T fragment and non-effect allele A fragment in SKOV3 and 293T cells. ∗P < 0.05, ∗∗P < 0.01, ∗∗∗P < 0.001. (B) The EMSA assay showed differences in the allele-specific protein binding capacity of rs7027650.

Discussion

In this study, we conducted targeted sequencing of 130 kb on the COL15A1 gene at the 9q22.33 region and subsequently validated the results in 1,099 EOC cases and 1,591 controls. We identified an unreported genetic variant, rs7027650, that was significantly associated with ovarian cancer risk and significantly correlated with the expression of COL15A1 in eQTL analysis. Preliminary molecular analysis revealed that rs7027650 may regulate the expression level of COL15A1.

In the previous three-stage GWAS study of ovarian cancer in Han Chinese women, we discovered two new susceptibility loci associated with the risk of epithelial ovarian cancer (Chen et al., 2014). SNP rs1413299 at the 9q22.33 region was the most significant SNP in this GWAS study. However, the functional variant at this locus and the molecular mechanism accounting for the risk of ovarian cancer were still unknown. Therefore, we performed fine-scale mapping, validation in a large case control cohort, and eQTL analysis to explore the candidate functional variants at the susceptibility risk loci 9q22.33. We determined that rs7027650 located at intron two of the COL15A1 gene was a potential candidate causal variant in this region.

Functional annotation revealed that rs7027650 overlaps with DNase I hypersensitivity sites in normal ovarian tissue and ovarian cancer cell lines, suggesting that this variant resides in open chromatin region and may serve a transcriptional regulation function. The Luciferase assay revealed that the gene fragments with rs7027650 have lower activity in SKOV3 and 293T cells indicating that the SNP could interact with the promoter region of COL15A1. The EMSA showed that rs7027650 exhibited different protein binding capacities between allele-A and allele-T. Collectively, these molecular analyses and experiments suggested that SNP rs7027650 may be a functional variant, potentially affecting COL15A1 transcription regulation and may affect ovarian cancer carcinogenesis.

The COL15A1 gene encodes the alpha chain of type COL15 which is a member of the FACIT collagen family. COL15 is a structurally complex macromolecule with many unique features and a range of biological properties. Light microscopy experiments have reported that COL15 is localized to endothelial, muscle, nerve, adipose, and most epithelial basement membrane regions of human tissues. This wide distribution suggests COL15 has an adhesive role, connecting the basement membrane to connective tissue (Amenta et al., 2005; Clementz & Harris, 2013). In breast cancer, collagen types XV and XIX were lost early in the development of invasive tumors. The disappearance of these proteins may signal remodeling of the extracellular matrix to promote tumor cell infiltration (Amenta et al., 2003). In a transgenic MMTV-PyMT mouse mammary carcinoma model, inactivation of COL15A1 modulated the tumor extracellular matrix and increased mammary tumor growth (Martinez-Nieto et al., 2021). Type XV collagen played a role in the adherence of the basement membrane to surrounding connective tissue and it may be associated with the tumorigenesis of keratinocytes and melanocytes (Fukushige et al., 2005). In cervical carcinoma cells, collagen XV functioned as a dose-dependent suppressor of tumorigenicity (Harris, Harris & Hollingsworth, 2007), and was found to inhibit the adhesion and migration of fibrosarcoma cells when present in fibronectin-containing matrices (Hurskainen et al., 2010). Kimura et al. (2016) reported that immunohistochemical examination of collagen XV and COL15A1 mRNA both showed increased expression in tumoral regions than in non-tumoral regions in human hepatocellular carcinoma (HCC), and collagen XV was considered to be the factor that contributed to the capillarization of HCC. Collagen XV was also reported to function as a metastasis inhibitor in HCC by regulating the discoidin domain receptor 1 (DDR1)-Snail/Slug axis, thus regulating epithelial-mesenchymal transition (EMT) (Yao et al., 2022). The COL15A1 gene has been reported to be upregulated by the interaction between Clusterin (CLU) and myocyte enhancer factor 2A (MEF2A). This upregulation is associated with blocking the EMT process, inhibiting the invasion and metastasis of testicular seminoma (Cui et al., 2021). In ovarian cancer, COL15A1 gene expression was significantly lower in tumor tissues than normal tissues in TCGA data (Chen et al., 2014), suggesting COL15A1 has a potential tumor suppressor role in ovarian cancer. Further experimental investigation is warranted to reveal the underlying molecular mechanism of COL15A1 in ovarian cancer carcinogenesis.

There are still some limitations to this study. Due to insufficient samples, we did not validate our results in another larger sample. We also did not find any significant association between rs7027650 and ovarian cancer risk in the European population from the Ovarian Cancer Association Consortium (OCAC) data. The inconsistent result with European population may be due to the race- and ethnicity heterogeneity of cancer susceptibility (Henderson et al., 2012). The ethnicity specific cancer susceptibility was also observed in several other kinds of cancers, such as colorectal cancer and prostate cancer etc. (Lu et al., 2019; Rebbeck, 2017). Additionally, this study only conducted a preliminary exploration of the potential function of the causal variant, and more experiments are needed to provide additional evidence to support the regulatory function of this variant.

Conclusions

In conclusion, we identified a potential causal variant within the chromosome 9q22.33 region, rs7027650, which was associated with ovarian cancer susceptibility. We also demonstrated that rs7027650 may regulate COL15A1 gene expression, offering suggestions for further functional studies to clarify the underlying biological mechanism in ovarian cancer etiology. Nonetheless, the transcriptional regulation mechanism of rs7027650 on COL15A1 remains unclear and needs to be clarified through further functional experiments.

Supplemental Information

Supplemental Information 1 LD structure around the target SNP rs1413299 at 9q22.33 based on HapMap Project database (CHB+JPT)

The high linkage disequilibrium between two variants is marked as a red square and low LD is marked as white.

Click here for additional data file.

Supplemental Information 2 The eQTL result of rs7027650 in ovarian cancer tumor tissues

Click here for additional data file.

Supplemental Information 3 Primers used in this study

Click here for additional data file.

Supplemental Information 4 Candidate causal variants selected for validation study from target sequencing

MAF, Minor allele frequency; SNP, singlenucleotide polymorphism. a Position is GRCh37. b Major allele/minor allele. c r2 of linkage disequilibrium between variants with rs1413299.

Click here for additional data file.

Supplemental Information 5 Candidate causal variants selected for validation study from the discovery stage of previous GWAS study a

GWAS, Genome-wide association study; MAF, minor allele frequencey; OR, odds ratio; SNP, single nucleotide polymorphism. a GWAS stage I in Han Chinese (1,172 controls/1,044 cases); genotyping using Illumina HumanOmniZhongHua-8 BeadChip. b Position is GRCh37. c Minor allele/major allele. d P value of association analysis using logistic regression adjusted for age and first three principal components of population stratification.

Click here for additional data file.

Supplemental Information 6 Candidate causal variants selected for validation study through SNAP (high LD with rs1413299)

a Position is GRCh37. b Risk allele/other allele. c r2 of linkage disequilibrium between variants with rs1413299.

Click here for additional data file.

Supplemental Information 7 Demographic characteristics of the participants

a: mean ±s.d.

Click here for additional data file.

Supplemental Information 8 Association results of ten SNPs at 9q22.33 in validation study

CI, confidence interval; OR, odds ratio; RAF, risk allele frequency; SNP, single nucleotide polymorphism. a Position is given with respect to genome build 37. b Risk allele/other allele.

Click here for additional data file.

Supplemental Information 9 Association results of ten SNPs stratified by histologic subtypes in validation study

CI, confidence interval; OR, odds ratio; RAF, risk allele frequency; SNP, single nucleotide polymorphism. a Position is given with respect to genome build 37. b Risk allele/other allele

Click here for additional data file.

Supplemental Information 10 The eQTL results of candidate causal variants from GTEx data

Click here for additional data file.

Supplemental Information 11 Raw data of validation dataset genotype result

Click here for additional data file.

Supplemental Information 12 Raw data of eQTL analysis

Click here for additional data file.

Supplemental Information 13 MIQE checklist

Click here for additional data file.

Additional Information and Declarations

Competing Interests

Author Contributions

Human Ethics

Data Availability

The authors declare there are no competing interests.

Tongyu Xing analyzed the data, prepared figures and/or tables, authored or reviewed drafts of the article, and approved the final draft.

Yanrui Zhao conceived and designed the experiments, analyzed the data, prepared figures and/or tables, and approved the final draft.

Lili Wang performed the experiments, prepared figures and/or tables, and approved the final draft.

Wei Geng performed the experiments, prepared figures and/or tables, and approved the final draft.

Wei Liu analyzed the data, prepared figures and/or tables, and approved the final draft.

Jingjing Zhou analyzed the data, prepared figures and/or tables, and approved the final draft.

Caiyun Huang performed the experiments, prepared figures and/or tables, and approved the final draft.

Wei Wang performed the experiments, prepared figures and/or tables, and approved the final draft.

Xinlei Chu performed the experiments, prepared figures and/or tables, and approved the final draft.

Ben Liu performed the experiments, authored or reviewed drafts of the article, and approved the final draft.

Kexin Chen conceived and designed the experiments, authored or reviewed drafts of the article, and approved the final draft.

Hong Zheng conceived and designed the experiments, authored or reviewed drafts of the article, and approved the final draft.

Lian Li conceived and designed the experiments, authored or reviewed drafts of the article, and approved the final draft.

The following information was supplied relating to ethical approvals (i.e., approving body and any reference numbers):

Medical Ethics Committee of Tianjin Medical University Cancer Institute and Hospital

The following information was supplied regarding data availability:

The raw measurements are available in the Supplementary Files.

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
