# Peer review of "Fine-scale mapping of chromosome 9q22.33 identifies candidate causal variant in ovarian cancer"

_PeerJ, doi:10.7717/peerj.16918_

## Round 0.1 · original submission · Major Revisions

The manuscript has been reviewed by 2 experts in this field, who have each provided comments and criticisms of the work. Most importantly, you will need to answer the concerns raised by reviewer 1, concerning the samples you used and the disparity of your results with the data from OCAC.

**Language Note:** The review process has identified that the English language must be improved. PeerJ can provide language editing services - please contact us at copyediting@peerj.com for pricing (be sure to provide your manuscript number and title). Alternatively, you should make your own arrangements to improve the language quality and provide details in your response letter. – PeerJ Staff

Reviewer 1 ·

Basic reporting

The authors report a fine mapping study of the region 9q22.33 in ovarian cancer in Han Chinese using 1099 cases and nearly 1600 controls. They report 2 new significant SNPs in the region rs7027650 (P=1.91×10-7, OR=1.31, 95%CI=1.21-158 1.41) and rs1889268 (P=3.71×10-2, OR=1.14, 95%CI=1.02-1.26). The original SNP from 2014 rs1413298 lost its overall association with OC although was still specifically associated with Endometrioid OC (P=8.72×10-3). The authors do not make it clear whether these were 'new' ovarian cancer samples or whether there was overlap with their 2014 paper. This is important especially regarding loss of association with rs1413298. The authors also provide evidence that rs7027650 in intron 6 of COL15A1 may affect expression of these gene but do not provide evidence that this protein affects ovarian cancer risk. The main problem with this paper is that freely available data from the Ovarian Cancer Association Consortium (OCAC) effectively refutes any association in white Europeans of both rs7027650 and rs1413298 with ORs of 1.0 and tight 95%CI of 0.98-1.02 and this in particular means that these cannot be functional SNPs for ovarian cancer risk. The authors must discuss the lack of validation in other populations and how they would still claim any real association for this region. rs1413298 actually had a negative association with endometrioid OR=0.969

Experimental design

This is appropriate although it is not clear why they have less OC samples than in their original 2014 paper (1,057 EOC cases and 1,191 controls in stage I, and replicate 41 SNPs (P(meta)<10(-4)) in 960 EOC cases and 1,799 controls (stage II), and an additional 492 EOC cases and 1,004 controls (stage III).) 1057+960+492 =2509 samples yet the authors claim they only had 1099 EOC cases with no possibility of a validation. The authors must state why these samples were no longer available and what the overlap was with this original sample set

Validity of the findings

The main problem here is that there has been no validation of their 9q22.33 region since 2014 apart from this paper. For them to claim that rs7027650 is a functional SNP this would be evident in other populations it is not. Freely available OCAC data https://ocac.ccge.medschl.cam.ac.uk/data-projects/results-lookup-by-region/ show that rs7027650 which has a position of 101741969 on build 37 and 98979687 on build 38 has an odds ratio of 1.00 with a CI of 0.98-1.02 in the HRC data set. This is a much larger collaboration and effectively refutes the authors main findings

·

Basic reporting

The manuscript is clearly written, however the English needs to improve. Sufficient references, background, tables and figures.

Experimental design

The study has a well defined research question, and the study is within the aims and scope of the journal and fulfil requirements of high technical and ethical standard. Methods well described overall.

Validity of the findings

1. Row 186, ¨significant association was not observed in normal ovary tissue¨, in Table S8 the only normal tissue tested for the marker rs7027650 was fibroblasts and not normal ovary as stated - and it was according to the table stat.significant? (variant id lacking rs- here). Could this perhaps suggest that there was no support for functional effect from the eQTL results?

Additional comments

This is an interesting and well conducted study presenting several lines of support for one candidate SNP, rs7027650 to act as a functional risk-allele in ovarian cancer

---

## Round 0.2 · Minor Revisions

Please see the reviewer's comment and respond. This may be as simple as adding a sentence to the Discussion to acknowledge the issue.

If you can acknowledge this, there will be no need for re-review.

Reviewer 1 ·

Basic reporting

Good

Experimental design

Good

Validity of the findings

I am still concerned about the suggestion that rs7027650 is a functional SNP. I find it hard to believe this could by ONLY functional in Han Chinese. This needs further clarification

---

## Round 0.3 · accepted · Accept

Thank you for addressing the reviewer's concerns in a positive manner with additional data. My decision is therefore to accept the paper without needing further review comments. The manuscript may need light editing for language but is otherwise acceptable for publication.